# A Genetic Window on Sardinian Native Horse Breeds through Uniparental Molecular Systems

**DOI:** 10.3390/ani10091544

**Published:** 2020-09-01

**Authors:** Andrea Giontella, Irene Cardinali, Camillo Pieramati, Raffaele Cherchi, Giovanni Paolo Biggio, Alessandro Achilli, Maurizio Silvestrelli, Hovirag Lancioni

**Affiliations:** 1Department of Veterinary Medicine, University of Perugia, 06123 Perugia, Italy; andrea.giontella@unipg.it (A.G.); camillo.pieramati@unipg.it (C.P.); maurizio.silvestrelli@unipg.it (M.S.); 2Department of Chemistry, Biology and Biotechnology, University of Perugia, 06123 Perugia, Italy; hovirag.lancioni@unipg.it; 3AGRIS, Servizio Ricerca Qualità e Valorizzazione delle Produzioni Equine, piazza D. Borgia, 4, Ozieri, 07014 Sassari, Italy; rcherchi@agrisricerca.it (R.C.); giampaolo.biggio@tiscali.it (G.P.B.); 4Department of Biology and Biotechnology “L. Spallanzani”, University of Pavia, 27100 Pavia, Italy; alessandro.achilli@unipv.it

**Keywords:** native horse breed, uniparental markers, horse phylogeny

## Abstract

**Simple Summary:**

The horse is a mammalian species showing a high variation among maternal lineages but a limited variability in the paternal inheritance. The female phylogenetic history is commonly investigated by analyzing the maternally transmitted mitochondrial DNA (mtDNA), while the male perspective is provided by the paternally inherited portion of the Y chromosome (NRY). Here we explored the variation of both non-recombining genetic systems in three horse breeds present in Sardinia: Giara, Sarcidano, and Sardinian Anglo-Arab. The analysis of 34 stallions revealed three differentially distributed NRY types: (i) the first and most ancestral one is typical of Sarcidano; (ii) the second is well represented in Giara and seems to derive from Neapolitan/Oriental stallions; (iii) the third confirms the Thoroughbred influence in the Sardinian Anglo-Arab breed. By extending the analysis to 178 mtDNAs, we observed a common maternal origin for Giara and Sarcidano. Contrarily, the outlier behavior of the Sardinian Anglo-Arab is due to its higher mitochondrial variability, testifying for multiple maternal lineages in its current population. Our preliminary findings highlight the importance of a parallel molecular screening of NRYs and mtDNAs to reconstruct both paternal and maternal phylogenetic histories and to fully evaluate the extent of autochthonous genetic resources in the island.

**Abstract:**

Sardinia, an island located to the west of Italy in the Mediterranean Sea, boasts three native horse breeds: Giara, Sarcidano, and Sardinian Anglo-Arab. Here, we have investigated for the first time three loci of the non-recombining region of the Y chromosome (NRY) in 34 stallions from these breeds and performed a phylogenetic analysis of the maternal relationships among 178 previously published mitochondrial control regions. We found that the current NRY diversity of Sardinian horse breeds is linked to three haplotypes (HT), all identified within Sarcidano. Each breed showed a typical HT: HT1 (ancestral) was the most represented in Sarcidano, HT2 (Neapolitan/Oriental wave) in Giara, and HT3 (Thoroughbred wave) in Sardinian Anglo-Arab. The specificity of each haplotype suggests the influence of independent breeding strategies and the effect of genetic drift in each Sardinian population. The female counterpart, extended to 178 horses, showed a low genetic variability and a common maternal origin for Giara and Sarcidano. The higher variability of the Sardinian Anglo-Arab indicates multiple mare lineages in its current population. Further genetic analyses will be crucial to understand the paternal history of male horses, preserve the endangered mares’ and stallions’ lineages, and improve the enhancement of autochthonous genetic resources on this island.

## 1. Introduction

Sardinia, located to the west of Italy in the Mediterranean Sea, is an island characterized by many autochthonous animal species. Among them, there are three horse breeds native to this island that are classified “at risk” by FAO (http://www.fao.org/dad-is) if considering their local status.

The Giara horse, with its population size of 524 individuals (298 females and 226 males) (FAO 2018; last update: August 2019), is one of the 15 indigenous horse “breeds of limited distribution” in Italy recognized by the AIA, the Italian breeders’ association. The origin of these animals is shrouded in mystery. Even if it is considered an ancient native breed, some scholars claim that they were introduced from the Middle East. In 1540, Sigismondo Arquer was the first to state that there were these wild horses in Sardinia [1]. In the XVIII century, numerous wild herds of Giara were observed in Sardinia and described in several reports. It takes the name from the Giara plateau, an area located at the Sardinian hinterland of about 45 km^2^ between 500 and 600 m above sea level. A mountain chain surrounds the plateau limiting any connections with the underlying valleys throughout time, thus preventing the migration of horses and protecting the biodiversity of this population. The breed is well adapted to dry and harsh environments where it lives in the wild.

The Sarcidano Horse is a rare semi-feral breed originated from the plateau of Sarcidano in the Laconi municipality (province of Oristano). It is also recognized by the AIA (http://www.aia.it) as “of limited distribution”, with 115 animals (65 females and 50 males) (FAO 2018; last update: March 2020). With the Giara breed, their local status is critical. They are both bred in semi-feral conditions and recorded in the Anagraphic Register (http://www.anagrafeequidi.it).

The Sardinian Anglo-Arab breed is the result of selected crossbreeding between indigenous Sardinian mares carrying Arabian blood and Thoroughbred and Arab stallions [2,3]. It is considered an endangered horse breed currently counting 3005 animals (2871 females and 134 males) (FAO 2018; last update: September 2019). The breed is maintained under controlled breeding conditions and its genealogies have been registered in a dedicated Studbook since 1967. The Sardinian Anglo-Arab is widely used in different equestrian competitions such as flat races and jumping [2,4]; for this reason, it was frequently crossbred with Thoroughbreds.

The horse is the typical domestic mammalian species that was selectively bred for specific traits. Few stallions, producing most desirable offspring, were often crossed with local autochthonous mares resulting in a high variation among the maternal lines but a limited variability in the paternal counterpart, both at macro- and micro-geographic levels. This breeding management is reflected in the uniparental markers acquired from female (mitochondrial DNA) and male (Y chromosome) ancestors [5,6,7,8,9,10,11,12]. The striking low diversity displayed by the non-recombining region of the Y chromosome (NRY) reflects the small number of stallions that contributed to the establishment of current breeds [6,10]. Recently, Wallner and colleagues described the nucleotide variation of modern horse Y chromosomes, resulting in six haplotypes distinct from the Przewalski horse, and traced their distribution using pedigree information [6,13].

On the other hand, the female counterpart of modern horses usually manifests a high genetic variability. In particular, the mitochondrial DNA (mtDNA) control regions of Sardinian horse breeds were already analyzed together with other Italian breeds, highlighting a genetic structure for these breeds, who live in geographically isolated contexts [7,14]. Previous studies based on shorter mtDNA control-region sequences claimed that Giara and Sarcidano consist of two distinct gene pools without gene flows between them [15]. The Giara horse was also described as genetically homogeneous and grouping in a unique clade with three ancient breeds (Sorraia, Garrano, and Potoka) [16].

The aim of this study was to investigate and reconstruct the paternal and maternal phylogenetic histories of three Sardinian horse breeds (Giara, Sarcidano, and Sardinian Anglo-Arab) (Table 1) by screening for the first time three loci of the NRY (YE3, YE17, and YXX) and by performing a parallel phylogenetic analysis on the mtDNA. This comparative study will shed light on their genetic peculiarities and evaluate the distribution of uniparental markers in present-day Sardinia, thus providing new knowledge for the enhancement of autochthonous genetic resources.

## 2. Materials and Methods

The analyses were conducted on a dataset of Sardinian horse DNAs previously collected and reported also in Cardinali et al., 2016 [7] and Giontella et al., 2020 [14]. All experimental procedures were reviewed and approved by the Animal Research Ethics Committee of the University of Perugia (Prot. N.2017_01).

For the Sardinian Anglo-Arab, genealogical information recorded in the Studbook was considered in order to select maternally unrelated samples belonging to dam lines with at least 100 descendants and deriving from all important and widespread sire lines. For the Giara and Sarcidano breeds, living in semi-feral conditions, sampling was performed during the late summer when the drought forces herds to reach the few available springs located within confined areas. These are equipped with corridors or small capture paddocks for the identification of subjects through a transponder, or any health treatments and biological sampling. In these peculiar circumstances, individuals were randomly sampled taking care to select specimens belonging to different groups in order to reduce the probability of sampling the genealogically closest subjects. Groups mainly consist of: solitary subjects (without herd); herds with a dominant stallion, about 8–10 mares and young foals; herds of only sexually mature young males.

Our complete dataset resulted in a total of 125 DNA samples (39 Giara, 30 Sarcidano, and 56 Sardinian Anglo-Arab) including 34 stallions.

### 2.1. Y-Chromosome Analysis

Three polymorphic sites of the NRY (YE3, YE17, and YXX) were analyzed for the 34 stallions belonging to the Sardinian breeds (12 Giara, 10 Sarcidano, and 12 Sardinian Anglo-Arab) as reported in Wallner and colleagues [6] (Appendix A).

Referring to Wallner et al., 2013 [6], we selected three pairs of oligonucleotides to amplify the polymorphic sites. For the locus YE3, we selected the forward primer (5′-CCCTCTGCTGAGCATCTAGG-3′) used to detect the mutation 10594delT and the reverse primer (5′-GGCTTAGGCCACTGATGGTA-3′) used to analyze the 20 single nucleotide polymorphisms (SNPs) and the indels. For the locus YE17, we amplified through the forward primer 5′-GGCCTAAGTTGTTCGCAGAG-3′ and the reverse 5′-TGACTGGTGGTGTCCAGTGT-3′, while to detect the SNP (Single Nucleotide Polymorphism) G/A in the locus YXX, we amplified by using the forward primer 5′-CCTCCGGCCTTTATGTCTTAG-3′ and the reverse 5′-TTGGGCTGCAGTATACAACG-3′. PCR reactions contained 1X Buffer GoTaq, 2.5 mM of each dNTPs, 0.3 μM of each primer, 0.03 U/μL of GoTaq DNA polymerase (Promega Corporation; Madison, WI, USA), 30 ng of genomic DNA, and H2O to a final volume of 25 μL. PCR amplification was carried out as follows: 95 °C for 2 min, followed by 35 cycles of 95 °C for 30 s, 62 °C for 30 s, 72 °C for 45 s, and then 72 °C for 5 min. The PCR fragments were purified using exonuclease I and alkaline phosphatase (ExoSAP-IT enzymatic system-USB Corporation, Cleveland, OH, USA) and subsequently Sanger-sequenced with the forward primer 5′-GCCAAACTACTCACCAGAAA-3′ for the locus YE3, 5′-GATTACCTCCTGGGACAAC-3′ for the locus YE17, and 5′-TAAAAACCTGTGGAAGGATAA-3′ for the locus YXX. Sequences were, respectively, aligned to the *Equus caballus* haplotype HT1 Y chromosome locus YE3 (JX646942.1), locus YE17 (JX646950.1), and locus YXX (JX647030.1) for the haplotype annotation, and their evolutionary relationships were evaluated through a median-joining tree built using Network software v.10.0 [17].

### 2.2. Mitochondrial DNA Control-Region Analysis

To provide a more comprehensive overview of the autochthonous horse breeds in Sardinia, the DnaSP 5.1 software [18] was used to estimate the haplotype variability of the mitochondrial control-region sequences reported in Cardinali et al., 2016 [7] and Giontella et al., 2020 [14], belonging to Giara (GI; *N* = 39), Sarcidano (SC; *N* = 30), and Sardinian Anglo-Arab (AA; *N* = 56) (Appendix A). Then, we evaluated the evolutionary relationships among the three breeds through the Network software v.10.0 [17], by including also the other Giara (*N* = 30) and Sarcidano (*N* = 23) sequences available from GenBank [15,16] (Appendix A).

## 3. Results

### 3.1. Y-Chromosome Analysis

The alignment of the 34 NRY sequences to the published reference sequences (JX646942.1, JX646950.1 and JX647030.1), from the nucleotide position (np) 10,592 to np 11,330 for the locus YE3, from nps 1240 to 1410 for YE17, and from nps 25,342 to 25,480 for YXX, showed three different haplotypes (HT1, HT2, and HT3) [6] (Appendix A). None of the samples presented mutations at the locus YXX. Considering all three loci, no mutations (HT1) have been identified in eight samples (three Giara and five Sarcidano). The HT2, characterized by a single nucleotide mutation (1277A), was found in 15 samples (nine Giara, one Sarcidano, and five Sardinian Anglo-Arab), while the HT3, differentiated from HT2 by a deletion in np 10,594, was found in 11 stallions (four Sarcidano and seven Sardinian Anglo-Arab) (Figure 1).

The Sarcidano is the only breed showing all three haplotypes. On the other hand, each breed is characterized by a most represented HT: HT1 in Sarcidano, HT2 in Giara, and HT3 in Sardinian Anglo-Arab (Figure 2).

### 3.2. Mitochondrial DNA Control-Region Analysis

The overall alignment of 610 base pairs, from np 15,491 to np 16,100, of the 125 control-region sequences from Cardinali et al., 2016 [7] and Giontella et al., 2020 [14] showed a haplotype diversity (Hd) of 0.944, with a total of 38 haplotypes and 49 polymorphic sites (S) detected (Table 2).

We observed similar values for Giara and Sarcidano in terms of haplotype diversity (0.883 and 0.828, respectively) and number of haplotypes (Nh; nine and eight, respectively), and higher values for Sardinian Anglo-Arab (Hd = 0.969; Nh = 30).

By analyzing a shorter mtDNA control region (from np 15,491 to np 15,740), in order to add another 53 published sequences from Giara and Sarcidano, we observed a total of 47 haplotypes, of which nine were shared between two or more Sardinian breeds while the other 38 were present in only one breed: 11 for Giara, 10 for Sarcidano, and 17 in Sardinian Anglo-Arab (Figure 3).

All haplotypes were classified in haplogroups, except for three sequences (JF804119, JF804147, and JF804151) from Morelli et al., 2014 [15], and a total of eight (A, B, E, G, I, L, M, and N) mtDNA lineages were identified, with the highest frequencies for G, L, and I (37%, 24%, and 21%, respectively) (Appendix A). The highest maternal variability was found in Sardinian Anglo-Arab, which showed all eight haplogroups, followed by Giara, with six mtDNA lineages, and then by Sarcidano, presenting only four haplogroups (Figure 4).

## 4. Discussion

Even if whole-genome approaches are now opening up new clues on the livestock genetic complexity and admixture, the two uniparental systems are still widely used to solve questions about breed origins and demographic processes. The traceability and characterization of female and male livestock lineages also offer a unique opportunity to conserve genetic resources and to promote and defend local breeds [19,20,21]. This approach is compelling for horses [9,22] and even more important for animals that are bred in semi-feral conditions when few genealogical data are available. In this perspective, centers for breed preservation (in situ ed ex situ) have been established for Giara and Sarcidano, where individuals genetically characterized for both parental lines and specifically selected are kept for the management and conservation of local genetic resources.

### 4.1. Y-Chromosome Analysis

The current Y-chromosomal diversity of Sardinian horse breeds is linked to three haplotypes (HT1-3) that have been previously identified, at relatively high frequencies, across a broad range of horse breeds [6]. The ancestral HT1 was found in Giara and Sarcidano horses, while HT2, considered a marker of the Neapolitan/Oriental wave, was present in all three Sardinian breeds. Finally, HT3, representing the Thoroughbred wave, was found in Sarcidano and Sardinian Anglo-Arab. Among our samples, the most frequent Y-chromosomal haplotype was HT2 (44%), whose distribution in Central Europe is in agreement with the spreading of Neapolitan and Oriental stallions from the Middle East to Central and Western Europe [6]. Despite the fact that Sarcidano was the only Sardinian breed presenting all three haplotypes, it had the lowest frequency for HT2 (10%).

The presence of only six haplotypes in European horses and the low microsatellite variability [6,23,24,25] testify for a decline of the male effective population size and for the consequent decrease of the NRY variability [13]. The practice of importing stallions (Arab and Thoroughbred) from foreign countries to improve local herds often caused the complete replacement of autochthonous Y chromosomes [13]. This could be observed also in our dataset, where each breed is characterized by distinctive haplotype frequencies that probably highlight the consequences of three paternal introgressions of imported stallions into local breeds (Figure 2). Giara shows a high percentage (75%) of HT2, frequent in Central and Eastern Europe, but absent in the breeds from Northern Europe and Iberian Peninsula, thus confirming genetic traces of horses with Middle Eastern origins, as reported by Gratani [26]. The Sardinian Anglo-Arab shows the highest frequency (58%) for HT3, typical of Thoroughbred (derived from Oriental lineages and responsible for the predominance of this haplotype in modern horses) and distributed across many warm-blood horse breeds. This prevalence confirms that the Sardinian Anglo-Arab is a breed involved in breeding programs to improve the sport aptitude [27]. The prevalence (50%) of the ancestral haplotype HT1 in Sarcidano confirms its uncertain (perhaps ancient) origins.

### 4.2. Mitochondrial DNA Control-Region Analysis

The unexpected high value of the overall haplotype diversity in Sardinian breeds is mostly driven by the Sardinian Anglo-Arab that presents 31 of the 38 haplotypes identified in our dataset (Table 2). The lower number of haplotypes identified in Giara and Sarcidano (Nh = 9 and 8, respectively) is widely shared among these breeds, thus confirming the geographically isolated context of the island, as previously reported [7,15], and indicating common maternal local sources.

To include all available data, for a total of 178 horses, we performed the network analysis after sequence trimming to make the mtDNA dataset homogeneous. The analysis revealed a total of 47 haplotypes, with 38 unique mtDNA control-region sequences (Figure 3). Among the other nine haplotypes, two were shared between Giara and Sarcidano horses, three were present only in Giara and Sardinian Anglo-Arab, and four were shared among the three breeds, testifying for a marginal impact of the latter breed on the original mtDNA gene pool of Giara and Sarcidano.

The haplogroup classification of all Sardinian sequences showed a total of eight horse maternal lineages out of the 17 previously reported [5] with one main lineage for each breed (Figure 4): haplogroup G mostly identified in Giara horses (68%), haplogroup I mainly represented in Sarcidano (41%), and haplogroup L, frequent in Sardinian Anglo-Arab (41%) (Appendix A). G and I haplogroups are more common in Asia and the Middle East, respectively [5]. The percentage of L in Sardinian Anglo-Arab is considerably higher than that recorded in Western Asia (18%) and Continental Europe (31%) [7].

### 4.3. Combining Information from Both Uniparental Genetic Systems

The Y-chromosome HT2 and the high number of maternally inherited haplotypes identified in the Sardinian Anglo-Arab breed suggest a sex-bias in the breeding program, where probably stallions from a unique breed (Thoroughbred) have been crossed with mares of different origins, a common breeding procedure used to improve local breeds with Thoroughbred thus creating half-breed individuals. The high mtDNA variability found in the Sardinian Anglo-Arab could be explained by the ancient migratory events that in the past have reached Sardinia and brought mares of different origins (especially old Arabian lines and Part-Arab) [14].

The semi-feral Sarcidano breed was probably less impacted by these breeding programs and conserved the highest NRY variability and strong traces of the ancestral haplotype HT1. Likewise, on the maternal side, it shows the four most common mtDNA haplogroups in current Italian breeds [7].

In Giara the high incidence of the mtDNA haplogroup G together with the results obtained from the NRY analysis confirms the Middle Eastern origin of this breed and its subsequent isolation in the island. Contemporary Giara horses are probably descendants of the ancient Phoenicia, in agreement with Gratani [26] and many non-official records stating that these horses were introduced in Sardinia by Phoenician in the first millennium BCE. Recent studies focusing on the genetic history of human Sardinian populations inferred extensive exchange and continuity between the Phoenician population and broader Sardinia [28,29], thus supporting the hypothesis that Phoenician sailors might have brought their horses during their maritime expansions across the Mediterranean Sea and established settlements along the southern shores of Sardinia. The Giara horses are now the only survivors of a race that until the Middle Ages was much more extensive. The isolated environment of the Giara plateau presumably contributed to preservation of the breed’s genetic structure.

## 5. Conclusions

Sardinia, located to the west of Italy in the Mediterranean Sea, is an island characterized by many autochthonous species. Among them, there are three horse breeds native to this island and classified “at risk”, which were the objective of the present study. The peculiar Y-chromosomal distribution together with the high frequencies of certain mtDNA haplogroups highlights the genetic characteristics of these endangered local breeds.

Each breed showed a distinctive NRY haplotype: the ancestral one (HT1), most represented in Sarcidano; the HT2, marker of the Neapolitan/Oriental wave, above all found in Giara, and HT3, legacy of the Thoroughbred wave, prominent in the Sardinian Anglo-Arab. As for the female counterpart, the haplotype sharing between Giara and Sarcidano supports inferences of common maternal origins and the high incidence of the four most frequent lineages in present-day Italian horse breeds (G, I, L, and M) suggests female inputs from the peninsula. However, Giara shows strong genetic relationships to ancient Eastern Mediterranean sources, corroborating the hypothesis of a common past migration traced back to Phoenicians: both humans and horses could have reached the island from the Eastern Mediterranean area by sea in the first millennium BCE.

The low variability in Giara and Sarcidano confirms the marginal impact of crossbreeding on the indigenous mitochondrial gene pools and the lack of recent gene flow from outside into Sardinia. On the other hand, the high haplotype variability reported for Sardinian Anglo-Arab indicates the presence of multiple mare lineages in the current population.

This study suggests that despite their endangered status, these Sardinian horse breeds preserve significant reservoirs of genetic diversity. Distinctive variants and/or lineages have been identified when evaluating the two uniparental genetic markers. Therefore, targeted conservation efforts that take into account both female and male lines of descent are required to safeguard the extant genetic variability and to establish improvements in the selection programs.

## Figures and Tables

**Figure 1 animals-10-01544-f001:**
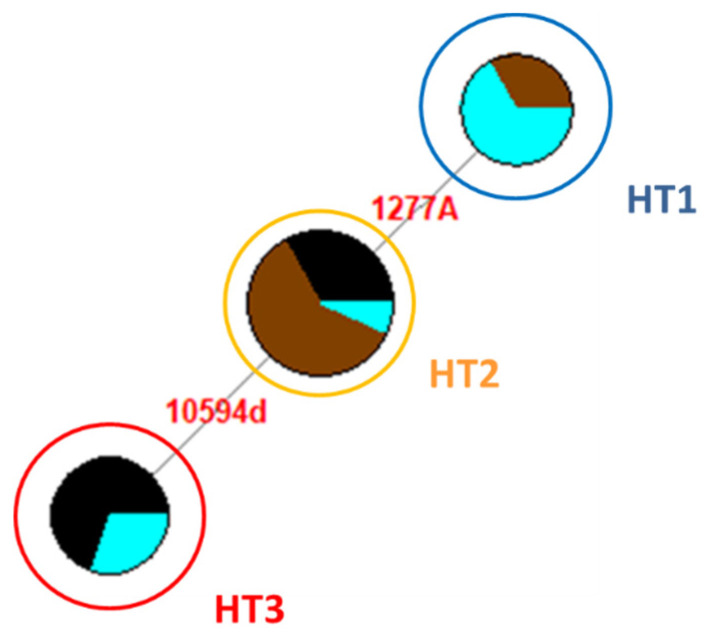
Median-Joining Network based on the non-recombining region of the Y chromosome (NRY) analysis of Sardinian horse breeds. Circles represent three different haplotypes and are proportional to the observed frequency in 34 Sardinian male horses (Appendix A). Colors refer to the three Sardinian breeds: Giara (brown), Sarcidano (light blue), and Sardinian Anglo-Arab (black).

**Figure 2 animals-10-01544-f002:**
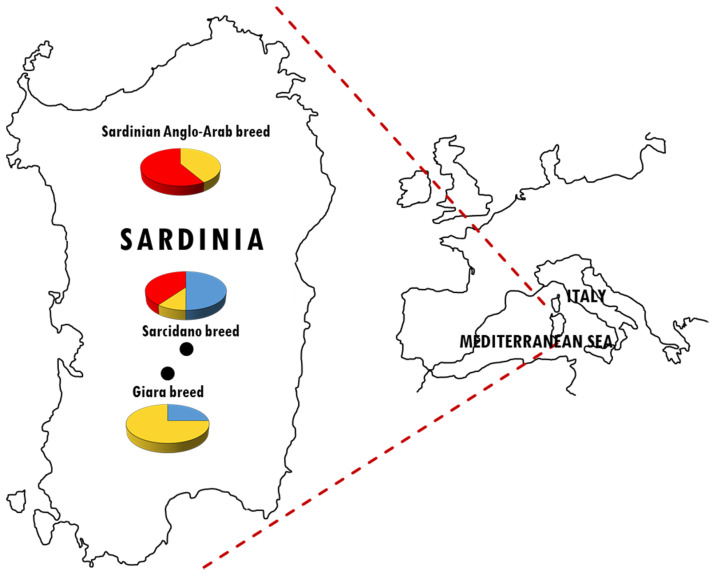
Sampling locations and Y-chromosomal haplotype frequencies of 34 stallions belonging to the Sardinian horse breeds. Colors refer to the different NRY haplotypes: HT1 (blue), HT2 (yellow), and HT3 (red).

**Figure 3 animals-10-01544-f003:**
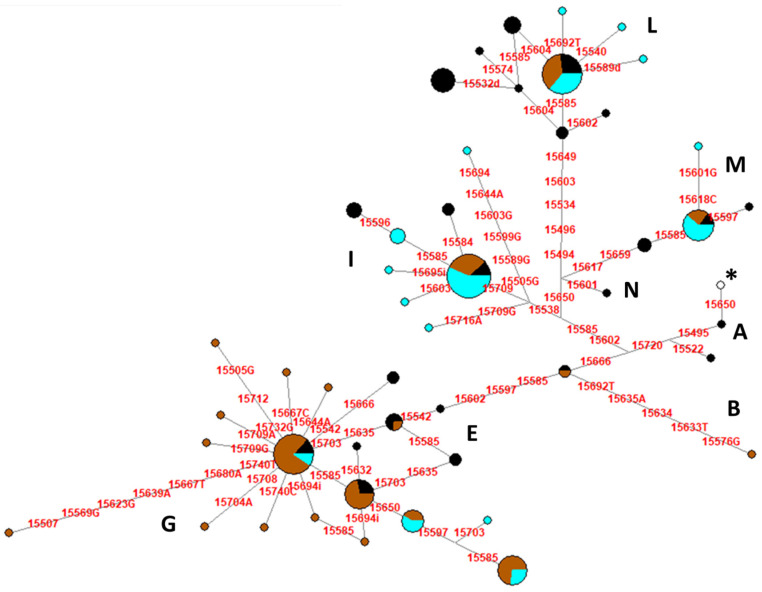
Median-Joining Network based on control-region sequences (nps 15,491–15,740) of Sardinian horse breeds. The asterisk indicates the haplotype identical to ERS. Three sequences not classified in mitochondrial DNA (mtDNA) haplogroups (JF804119, JF804147, and JF804151) were excluded from the analysis. Circles represent the haplotypes and are proportional to the observed frequency in 175 Sardinian horses (Appendix A). Capital letters refer to mtDNA haplogroups, while colors indicate the different breeds: Giara (brown), Sarcidano (light blue), and Sardinian Anglo-Arab (black).

**Figure 4 animals-10-01544-f004:**
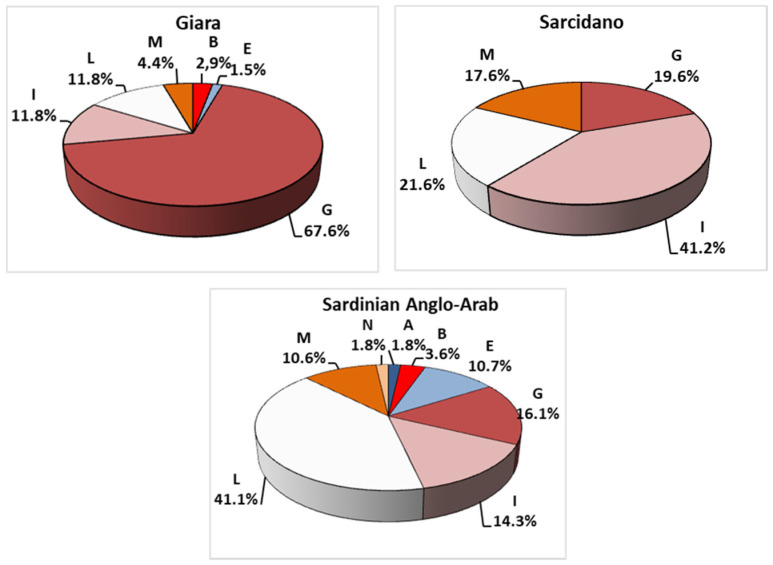
MtDNA haplogroup frequency distribution of 175 sequences belonging to the three Sardinian breeds available from GenBank. Three sequences not classified in mtDNA haplogroups (JF804119, JF804147, and JF804151) were excluded from the analysis.

**Table 1 animals-10-01544-t001:** List and current breeding context of Sardinian horse breeds.

Breed	Geographic Origin and Current Distribution	Breeding Conditions	Breed Composition ^a^	Genealogical Records	Classification According to Height at Withers	Uses	Temperament	Breed Classification (AIA) ^b^ (FAO) ^c^	
Sardinian Anglo-Arab	Sardinia; widespread	Controlled	3005 (2871 mares, 134 stallions)	Studbook, 1967	Horse	Racing, jumping, other equestrian disciplines	Lively, fiery, courageous	Autochthonous breed	National breed
Giara	Giara plateau; Sardinia South-Center	Semi-feral	524 (298 mares, 226 stallions)	Anagr. Reg., 1995	Pony	Work, country horse riding	Fiery, rustic, frugal	Autochthonous breed	Local breed (Endangered)
Sarcidano	Sarcidano plateau; Sardinia Center	Semi-feral	115 (65 mares, 50 stallions)	Anagr. Reg., 2006	Pony	Work, country horse riding	Lively, tame, adaptable to equestrian sports	Autochthonous breed	Local breed

^a^ Breed composition = number of individuals recorded in Studbooks or Anagraphic Registers, retrieved from FAO (http://www.fao.org/dad-is). Values for Giara and Sarcidano have to be considered underestimated because they reflect only subjects to whom it was possible to insert the microchip (transponder). ^b^ The Ministerial Decree n. 1598 of 23 January, 2015 reported the 15 breeds recorded in the Italian Registry of Autochthonous Equine Breeds. ^c^ Based on the establishment of Studbook or Anagraphic Register.

**Table 2 animals-10-01544-t002:** Genetic diversity indices.

Breed	Breed Code	Range	N	π	Nh	Hd	S	k
Giara	GI	nps 15,491–16,100	39	0.016	9	0.883	30	10.024
Sarcidano	SC	nps 15,491–16,100	30	0.017	8	0.828	30	10.561
Sardinian Anglo-Arab	AA	nps 15,491–16,100	56	0.019	31	0.969	49	11.836
Total		nps 15,491–16,100	125	0.019	38	0.944	49	11.717

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
