# Peer review of "A Genetic Window on Sardinian Native Horse Breeds through Uniparental Molecular Systems"

_animals, 2020, doi:10.3390/ani10091544_

Round 1

Reviewer 1 Report

A1-The current breeding situation of these breeds in a table could summarise! Breed, description -standard- of the breed (reference), year of description, year of the establishing stud book, number of mares, stallions, reference.

A2-A map of the geographical origin in Sardinia could also helps visualisation of the important fact about populations.

A3-The genealogical informations from the studbooks (if the studbooks are present) are necessary for the sampling! To identify the paternal lineages and after the sampling with complete representation are necessary in this small populations! More details are necessary about sampling!

19 lines > lineages

19 genetic history > phylogenetic history

20-21 „non-recombining genetic elements”  expression could helps in focusing on the most important genetic feature of these genetic elements. NRY abbeveration in Brem article is also non-recombining Y region!

31 genetic histories > phylogenetic histories (because we are investigated only the non recombinating genetic elements, and the „genetic histories” sounds a little bit wider than non recombinating genetic elements)

46  Maybe here is necessary aso to mention the „preservation selection” for supporting the preserving of the endangered mare and stalions lineages

66 AIA   and69 Anagraphic Register > missing the web adress of the description of this register!

74 dedicated Studbook > the establishing year of this sudbook, if it is given, could be very informative here!

94 paternal and maternal histories> paternal and maternal phylogenetic histories

98 of uniparental markers > markers?

105 Some words are necessary here about the sampling (e.g. random, not related individual, founder), the reference is not enough!

149 male horses> stallions

Figure 2. Maybe this pie-charts would be more informative and nicer if the diagrams would be merged with the geographic distribution on Sardinia map of breeds! Like in Brem article:

159 The Material is totally missing from here! The sampling proceducedure for mtDNA is essential in the study!

206 few genealogical data are available - the table is necessary about breeds data A1!

251-253 This breeding method were common for the improving the local breed with Thoroughbreed- halfbreed construction

254 impacted by the breeding program> impacted by these breeding program

293 in the conclusions maybe mentioned the „preservation selection” Imre Bodó (Conservation genetics of endangered horse breeds -  I. Bodo, L. Alderson, B. Langlois), what is able to preserve the diversity of Y and mtDNA haplotypes.

Author Response

A1-The current breeding situation of these breeds in a table could summarise! Breed, description -standard- of the breed (reference), year of description, year of the establishing stud book, number of mares, stallions, reference.

Authors’ response: Thank you for the suggestion, we have added a table summarizing all this information: “Table 1. List and current breeding context of Sardinian horse breeds.”

A2-A map of the geographical origin in Sardinia could also helps visualisation of the important fact about populations.

Authors’ response: Thank you for the suggestion, we have now modified the figure 2 by adding a map of the geographical origin for the three Sardinian breeds, thus changing the figure legend as follows: “Figure 2. Sampling locations and Y-chromosomal haplotype frequencies of 34 stallions belonging to the Sardinian horse breeds. Colors refer to the different NRY haplotypes: HT1 (blue), HT2 (yellow) and HT3 (red).”

A3-The genealogical informations from the studbooks (if the studbooks are present) are necessary for the sampling! To identify the paternal lineages and after the sampling with complete representation are necessary in this small populations! More details are necessary about sampling!

Authors’ response: We have appreciated the Reviewer’s suggestion and the requested information has been added in a paragraph at the beginning of Materials and methods section.

19 lines > lineages

Authors’ response: we have replaced "lines" with " lineages"

19 genetic history > phylogenetic history

Authors’ response: we have replaced "genetic history" with "phylogenetic history"

20-21 „non-recombining genetic elements”  expression could helps in focusing on the most important genetic feature of these genetic elements. NRY abbeveration in Brem article is alsonon-recombining Y region!

Authors’ response: Thank you, we have now introduced the suggested abbreviation throughout the text

31 genetic histories > phylogenetic histories (because we are investigated only the non recombinating genetic elements, and the „genetic histories” sounds a little bit wider than non recombinating genetic elements)

Authors’ response: Thank you, we have replaced "genetic history" with "phylogenetic history"

46  Maybe here is necessary aso to mention the „preservation selection” for supporting the preserving of the endangered mare and stalions lineages

Authors’ response: Thank you, we agree with the Reviewer and we have mentioned this concept by modifying the last sentence of Abstract as follows: “Further genetic analyses will be crucial to understand the paternal history of male horses, preserve the endangered mares and stallions’ lineages and improve the enhancement of autochthonous genetic resources in this island”

66 AIA   and69 Anagraphic Register > missing the web adress of the description of this register!

Authors’ response: we have added the web addresses

74 dedicated Studbook > the establishing year of this sudbook, if it is given, could be very informative here!

Authors’ response: we have added the establishing year of the Studbook

94 paternal and maternal histories> paternal and maternal phylogenetic histories

Authors’ response: Done

98 of uniparental markers > markers?

Authors’ response: We preferred to maintain the adjective in order to be more descriptive and precise

105 Some words are necessary here about the sampling (e.g. random, not related individual, founder), the reference is not enough!

Authors’ response: Thank you, we have now added the follow paragraph at the beginning of Material and Methods section: “The analyses were conducted on a dataset of Sardinian horse DNAs previously collected and reported also in Cardinali et al. 2016 [7] and Giontella et al. 2020 [14]. All experimental procedures were reviewed and approved by the Animal Research Ethics Committee of the University of Perugia (Prot. N.2017_01). For the Sardinian Anglo-Arab, genealogical information recorded in the Studbook was considered in order to select maternally unrelated samples belonging to dam lines with major consistency (with at least 100 descendants) and derived from all important and widespread sire lines. For the Giara and Sarcidano breeds, living in semi-feral conditions, sampling was performed during the late summer when the drought forces herds to reach the few available springs located within confined areas. These are equipped with corridors or small capture paddocks for the identification of subjects through a transponder, or any health treatments and biological sampling. In these peculiar circumstances, individuals were randomly sampled taking care to select specimens belonging to different groups in order to reduce the probability of sampling the genealogically closest subjects. Groups mainly consist in: solitary subjects (without herd); herds with a dominant stallion, about 8-10 mares and young foals; herds of only sexually mature young males. Our complete dataset resulted in a total of 125 DNA samples (39 Giara, 30 Sarcidano and 56 Sardinian Anglo-Arab) including 34 stallions.”

149 male horses> stallions

Authors’ response: Done

Figure 2. Maybe this pie-charts would be more informative and nicer if the diagrams would be merged with the geographic distribution on Sardinia map of breeds! Like in Brem article.

Authors’ response: Thank you for the suggestion, we have now modified the figure 2 by adding a map of the geographical origin for the three Sardinian breeds, thus changing the figure legend as follows: “Figure 2. Sampling locations and Y-chromosomal haplotype frequencies of 34 stallions belonging to the three Sardinian horse breeds here analysed. Colors refer to the different NRY haplotypes: HT1 (blue), HT2 (yellow) and HT3 (red).”

159 The Material is totally missing from here! The sampling proceducedure for mtDNA is essential in the study!

Authors’ response: Thank you, the sampling procedure was now reported at the beginning of Material and Methods section, as above reported in response to the comment for the line 105.

206 few genealogical data are available - the table is necessary about breeds data A1!

Authors’ response: Thank you for the suggestion, we have added a table summarizing all this information: “Table 1. List and current breeding context of Sardinian horse breeds.”

251-253 This breeding method were common for the improving the local breed with Thoroughbreed- halfbreed construction

Authors’ response: Thank you for the observation. Yes, this is a breeding procedure used to improve local breeds with Thoroughbred thus creating half-breed individuals. We specify this in the main text.

254 impacted by the breeding program> impacted by these breeding program

Authors’ response: Done

293 in the conclusions maybe mentioned the „preservation selection” Imre Bodó (Conservation genetics of endangered horse breeds -  I. Bodo, L. Alderson, B. Langlois), what is able to preserve the diversity of Y and mtDNA haplotypes.

Authors’ response: Thank you, we have now included the reference: Bodo I., Alderson L., Langlois B., editors. Conservation Genetics of Endangered Horse Breeds. 1st ed. Wageningen Academic Publishers; Wageningen, The Netherlands: 2005. However, we preferred not to include references in the conclusion paragraph and we included the reference in the discussion thinking that could be an appropriate citation in line 207

Reviewer 2 Report

The Y-chromosome analysis presented is interesting and provides new information regarding population structure of the 3 Sardinian native breeds. The mtDNA analysis for the longer fragment reuses sequences from 2 publications and does not appear to introduce critical new information or insight relative to Cardinali et al 2016. In my view, only the analysis of the shorter fragment, with additional sequences available from Genbank, should be presented in results. In the discussion about the importance of using molecular information for conservation purposes, could the authors propose how this could be accomplished for the 2 semi-feral breeds?

In addition, the following suggestions are offered to improve quality of writing and content.

Edits on lines:

61-63: A mountain chain surrounds the plateau which has limited… throughout time

64: … The breed is well …. in the wild.

73: … The breed is maintained under controlled …

77: … that was selectively bred for specific traits.

80: replace “impressed” with “reflected”.

89: … who “live” in ….

103-105: Clarify if samples from 34 stallions were also included in the mtDNA analysis. Based on sample ID (Table S2) this appears to be the case but should be disclosed in text.

160-177: Results for the longer mtDNA fragment appear to be a duplicate of Cardinali et al 2016. Authors need to explain what new results were produced, or delete this section.

189-190: Regarding the excluded 3 mtDNA sequences (one from Giara and 2 from Sarcidano), Morelli et al 2014 excluded them from their analysis because the polymorphisms didn’t fit with haplogroup definitions. For the current study, did the authors blast the 3 sequences to determine if similar sequences have been archived? If similarities exist, this should be discussed in terms of implications. These sequences could represent additional variation in these 2 breeds that, in my view, should not be dismissed, unless there are sourcing or quality problems. (I’m fully aware that these are Genbank sequences, not under the control of the authors).

192: “showed all eight”…

201: Replace “Even if” with “While…”

201: Replace “opening up” with “revealing”

215: Replace “Despite” with “While” and “it reaches” with “it has”

238: insert “mtDNA” before “analysis”

239: revise to “trimming to make a uniform data set.”

244: insert “maternal” before ‘lineages”

259: “descendants”

279: replace “testifies for” with “supports inferences of”

281: replace “might suggest” with “suggests” and “proves to preserve” with “shows”

Author Response

The Y-chromosome analysis presented is interesting and provides new information regarding population structure of the 3 Sardinian native breeds. The mtDNA analysis for the longer fragment reuses sequences from 2 publications and does not appear to introduce critical new information or insight relative to Cardinali et al 2016. In my view, only the analysis of the shorter fragment, with additional sequences available from Genbank, should be presented in results.

Authors’ response: Thank you for the suggestion. We decided to delete the network analysis for the longer mtDNA fragment and present only the phylogenetic relationships based on the shorter fragment, in order to include all the Sardinian horse sequences available from GenBank.

In the discussion about the importance of using molecular information for conservation purposes, could the authors propose how this could be accomplished for the 2 semi-feral breeds?

Authors’ response: Thank you for the interesting observation. We added the follow sentences: “In this perspective, centers for breed preservation (in situ ed ex situ) have been established for Giara and Sarcidano, where individuals genetically characterized for both parental lines and specifically selected are kept for the management and conservation of local genetic resources.”

In addition, the following suggestions are offered to improve quality of writing and content.

Edits on lines:

61-63: A mountain chain surrounds the plateau which has limited… throughout time

Authors’ response: Done

64: … The breed is well …. in the wild.

Authors’ response: Done

73: … The breed is maintained under controlled …

Authors’ response: Done

77: … that was selectively bred for specific traits.

Authors’ response: We have changed the sentence as suggested

80: replace “impressed” with “reflected”.

Authors’ response: We have replaced “impressed” with “reflected”.

89: … who “live” in ….

Authors’ response: Done

103-105: Clarify if samples from 34 stallions were also included in the mtDNA analysis. Based on sample ID (Table S2) this appears to be the case but should be disclosed in text.

Authors’ response: Thank you, we have now added a paragraph at the beginning of Material and Methods section, thus clarifying that the 34 stallions were included also in the mtDNA analyses: “… Our complete dataset resulted in a total of 125 DNA samples (39 Giara, 30 Sarcidano and 56 Sardinian Anglo-Arab), including 34 stallions.”

160-177: Results for the longer mtDNA fragment appear to be a duplicate of Cardinali et al 2016. Authors need to explain what new results were produced, or delete this section.

Authors’ response: Thank you. We decided to delete the network analysis for the longer mtDNA fragment.

189-190: Regarding the excluded 3 mtDNA sequences (one from Giara and 2 from Sarcidano), Morelli et al 2014 excluded them from their analysis because the polymorphisms didn’t fit with haplogroup definitions. For the current study, did the authors blast the 3 sequences to determine if similar sequences have been archived? If similarities exist, this should be discussed in terms of implications. These sequences could represent additional variation in these 2 breeds that, in my view, should not be dismissed, unless there are sourcing or quality problems. (I’m fully aware that these are Genbank sequences, not under the control of the authors).

Authors’ response: We performed the BLAST analysis for these three sequences and only one of them (JF804119) matched with other horse mtDNAs (five in total and all complete mitogenomes). For these reasons it was not possible to classify the three sequences from Morelli et al. 2014, and we decided to exclude them from our network analysis.

192: “showed all eight”…

Authors’ response: Done

201: Replace “Even if” with “While…”

Authors’ response: Thank you for the suggestion, but we prefer maintaining “even if”

201: Replace “opening up” with “revealing”

Authors’ response: Thank you for the suggestion, but we prefer maintaining “opening up”

215: Replace “Despite” with “While” and “it reaches” with “it has”

Authors’ response: Thank you for the suggestion, but we prefer maintaining “despite” and “it reaches”

238: insert “mtDNA” before “analysis”

Authors’ response: Done

239: revise to “trimming to make a uniform data set.”

Authors’ response: We have changed the sentence as suggested

244: insert “maternal” before ‘lineages”

Authors’ response: We have added “maternal”

259: “descendants”

Authors’ response: Done

279: replace “testifies for” with “supports inferences of”

Authors’ response: We have changed the sentence as suggested.

281: replace “might suggest” with “suggests” and “proves to preserve” with “shows”

Authors’ response: We have changed the sentence as suggested.

Reviewer 3 Report

The work is very interesting and brings important information to the subject of the protection of genetic resources of farm animals.

The work is very interesting and brings important information to the subject of the protection of genetic resources of farm animals. It is written in a transparent manner and does not raise any doubts in terms of its content. Needs minor corrections.

In section 5 "Discussion" there is no broader discussion of the mtDNS of Anglo-Arabo-Sardo horses, which show the greatest variability and the highest number of haplogroups (Fig. 5). Why is the mtDNA variability so high in this group of horses? As indicated by the authors, this breed was bred using imported stallions (Thoroughbred) mated with local female material. In the authors' opinion, where does the high number of maternally inherited haplotypes come from?

Author Response

The work is very interesting and brings important information to the subject of the protection of genetic resources of farm animals.

The work is very interesting and brings important information to the subject of the protection of genetic resources of farm animals. It is written in a transparent manner and does not raise any doubts in terms of its content. Needs minor corrections.

In section 5 "Discussion" there is no broader discussion of the mtDNS of Anglo-Arabo-Sardo horses, which show the greatest variability and the highest number of haplogroups (Fig. 5). Why is the mtDNA variability so high in this group of horses? As indicated by the authors, this breed was bred using imported stallions (Thoroughbred) mated with local female material. In the authors' opinion, where does the high number of maternally inherited haplotypes come from?

Authors’ response: Thank you for the observation. Sardinian Anglo-Arab horse breed originates from crosses between Sardinian indigenous mares with Arab and Thoroughbred stallions and this great variability in terms of mtDNA haplotypes and haplogroups highlights the presence of multiple mare lineages and is one of the higher among those observed in the continental Italy. This is probably due to some migratory events that in the past have brought mares of different origins into the island. We added the follow sentence in the Discussion paragraph: “The high mtDNA variability found in the Sardinian Anglo-Arab could be explained by the ancient migratory events that in the past have reached Sardinia and brought mares of different origins (especially old Arabian lines and Part-Arab).”
